# Fabricating Pea Protein Micro-Gel-Stabilized Pickering Emulsion as Saturated Fat Replacement in Ice Cream

**DOI:** 10.3390/foods13101511

**Published:** 2024-05-13

**Authors:** Xv Qin, Yaxian Guo, Xiaoqing Zhao, Bin Liang, Chanchan Sun, Xiulian Li, Changjian Ji

**Affiliations:** 1College of Life Sciences, Yantai University, Yantai 264005, China; 15063757029@163.com (X.Q.); gyx19991009@163.com (Y.G.); zxqytu1011@163.com (X.Z.); 2College of Food Engineering, Ludong University, Yantai 264025, China; 3School of Pharmacy, Binzhou Medical University, Yantai 264003, China; ouclixiulian@163.com; 4Department of Physics and Electronic Engineering, Qilu Normal University, Jinan 250200, China; jichangjiansdejcj@qlnu.edu.cn

**Keywords:** pea protein, micro-gel, O/W emulsion, saturated fat replacement, low-saturated fat ice cream

## Abstract

Unsaturated fat replacement should be used to reduce the use of saturated fat and *trans* fatty acids in the diet. In this study, pea protein micro-gels (PPMs) with different structures were prepared by microparticulation at pH 4.0–7.0 and named as PPM (pH 4.0), PPM (pH 4.5), PPM (pH 5.0), PPM (pH 5.5), PPM (pH 6.0), PPM (pH 6.5), and PPM (pH 7.0). Pea protein was used as a control to evaluate the structure and interfacial properties of PPMs by particle size distribution, Fourier transform infrared spectroscopy (FTIR), free sulfhydryl group content, and emulsifying property. PPM (pH 7.0) was suitable for application in O/W emulsion stabilization because of its proper particle size, more flexible structure, high emulsifying activity index (EAI) and emulsifying stability index (ESI). The Pickering emulsion stabilized by PPM (pH 7.0) had a uniform oil droplet distribution and similar rheological properties to cream, so it can be used as a saturated fat replacement in the manufacture of ice cream. Saturated fat was partially replaced at different levels of 0%, 20%, 40%, 60%, 80%, and 100%, which were respectively named as PR0, PR20, PR40, PR60, PR80, and PR100. The rheological properties, physicochemical indexes, and sensory properties of low-saturated fat ice cream show that PPM (pH 7.0)-stabilized emulsion can be used to substitute 60% cream to manufacture low-saturated fat ice cream that has high structural stability and similar melting properties, overrun, and sensory properties to PR0. The article shows that it is feasible to prepare low-saturated fat ice cream with PPM (pH 7.0)-stabilized Pickering emulsion, which can not only maintain the fatty acid profile of the corn oil used, but also possess a solid-like structure. Its application is of positive significance for the development of nutritious and healthy foods and the reduction of chronic disease incidence.

## 1. Introduction

Chronic diseases, such as cardiovascular disease, have become the most serious diseases threatening humans in the world today, seriously endangering people’s physical health. [1]. Many studies have shown that the intake of saturated fats and *trans* fatty acids increases the risk of chronic disease, whereas the intake of unsaturated fats decreases it [2,3]. Therefore, in recent years, the nutritional guidelines by various government agencies have recommended that unsaturated fat replacements could be used to reduce the use of saturated fat and *trans* fatty acids in the diet, so as to achieve a proportion of solid fat in total daily calorie intake of less than 10% [4].

Ice cream is a common oil-in-water system food. It is very popular among consumers because of its delicate, smooth taste and rich, mellow flavor. At present, ice cream processing generally continues the previous traditional ice cream processing methods, with water, milk, cream, sucrose, etc. as the main raw materials. It is characterized by high sugar, calories, and saturated fatty acid. Although consumers are already aware that excessive intake of saturated fat is not conducive to human health, ice cream requires saturated fat to provide structure and a creamy mouthfeel [5]. Therefore, researchers have begun to investigate different techniques for reducing saturated fat to ensure that healthier, nutritious foods are obtained without degrading or destroying the structural and organoleptic properties of the food [6].

Ice cream is a favorite casual dessert, and the market for low-saturated fat ice cream is growing rapidly as consumers focus on healthy eating. However, simply reducing the saturated fat content of ice cream can affect many properties such as viscosity, hardness, melting, and flavor. The difficulty in low-saturated fat ice cream is how to reduce the saturated fat content of ice cream without affecting its sensory properties [7]. Currently, the main approach to reduce saturated fat in food is to use saturated fat replacements (saturated fat mimics and saturated fat replacers like protein-based fat replacers, carbohydrate-based fat replacers, lipid-based fat replacers, and complex fat replacers) to substitute some or all of the saturated fat to ensure sensory properties while reducing the calories [8]. However, these saturated fat replacements have not been commercialized due to a number of problems. The ester bonds of saturated fat replacements cannot be hydrolyzed by lipases and therefore cannot be digested and absorbed by the body [9]. Thus, high intake of saturated fat replacements can lead to many intestinal problems. The application of protein matrix and carbohydrate matrix fat mimics would lead to loss of fat-soluble flavor substances, shortening the shelf life [10,11]. Therefore, the development of a novel saturated fat replacement is of positive significance for the development of nutritious and healthy foods.

In recent years, consumers have gradually realized that saturated fatty acid intake increases the incidence of chronic diseases, so the demand for low-saturated fat food increases year-by-year. Protein-stabilized O/W emulsions, including pea protein-, soybean protein-, wheat gliadin- and corn gliadin-stabilized emulsions, have attracted more and more attention as new saturated fat replacements [12]. On the one hand, protein-stabilized emulsions enable the solidification of vegetable oils. On the other hand, vegetable oil contained in plant protein-stabilized emulsions contains more unsaturated fat esters than fat, which is more nutritious [13]. However, protein-stabilized conventional emulsions are sensitive to environmental factors, such as acid and heat, and tend to destabilize during processing and storage [14]. Therefore, solving this destabilization problem by modifying the molecular structure of proteins is of increasing interest to the food industry.

The protein micro-gel particles produced by microparticulation require heating of the protein to form a gel structure while/later applying high shear to disrupt the gel network into particles [15]. Protein micro-gel particles tend to have great interfacial properties because of the increasing hydrophobicity of the protein surface during microparticulation [16]. Pickering emulsion stabilized by protein micro-gel particles has advantages such as good stability and biocompatibility [17]. Thus, we hypothesized that a protein micro-gel particle-stabilized Pickering emulsion could be used to replace saturated fat in ice cream.

Pea protein has antioxidant, anti-hypertension, anti-inflammatory, cholesterol reduction, and regulation of intestinal bacterial activity, and it has hypo-sensitization [18]. However, the poor solubility, emulsification, and emulsification stability of pea protein limit its application in food processing [19]. Thus, in this study, pea protein was microparticulated to improve its processing performance, such as emulsification. Different structural and physicochemical properties of pea protein micro-gel particles were obtained by adjusting the preparation pH during microparticulation. Then, an O/W emulsion stabilized by pea protein micro-gel particles was designed to substitute saturated fat and was used as saturated fat replacement in low-saturated fat ice cream. And the saturated fat in ice cream was partially replaced at different rates of 20%, 40%, 60%, 80%, and 100%. The physicochemical properties and rheological properties of full-saturated fat and low-saturated fat ice cream were investigated to find the most suitable replacement ratio.

## 2. Materials and Methods

### 2.1. Materials

Pea protein isolate (PPI, containing 85% protein) was obtained from Youcheng Biotechnology Co., Ltd. (Yantai, China). Corn oil was obtained from a local supermarket (Shandong, China). Sodium dodecyl sulfate was obtained from Shanghai MacLean Biochemical Technology Co., Ltd. (Shanghai, China). Skim milk powder was purchased from Dulbert Erie Dairy Co., LTD. Dilute cream was obtained from Qingdao Nestle Co., Ltd. (Qingdao, China) Anhydrous citric acid was obtained from Weifang Yingxuan Industrial Co., Ltd. (Weifang, China) Sodium alginate was obtained from Lianyungang Tiantian Seaweed Industry Co., Ltd. (Lianyungang, China) Sucralose was obtained from Shandong Sanhe Vicin Biotechnology Co., Ltd. (Tai’an, China)

### 2.2. Preparation of Pea Protein Micro-Gel Particles

PPI (18 g/100 mL) dispersion was magnetically stirred for 2 h, and the initial pH of PPI dispersion was 7.22. Then, the pH of PPI dispersion was adjusted from 4.0 to 7.0 using 1 M HCl solution. Then, the PPI dispersions at pH 4.0 to 7.0 were respectively heated at 90 °C for 30 min. The PPI gels were prepared by cooling the heated PPI dispersions to room temperature and then refrigerating at 4 °C for 12 h. Then, the PPI gels were homogenized in a T18 Ultra-Turrax (IKA, Staufen, Germany) at 8000 rpm for 3 min to obtain PPI micro-gel particles. They were named pea protein micro-gel (PPM) (pH 4.0), PPM (pH 5.5), PPM (pH 5.0), PPM (pH 6.0), PPM (pH 6.5), and PPM (pH 7.0), respectively.

### 2.3. Protein Structure Characteristics

#### 2.3.1. Particle Size Distribution of PPMs

Particle size distribution of the protein micro-gel was measured by a Battersize2000 particle size analyzer (Better, Dandong, China). The samples were added to a stirred measuring cell containing 800 mL of deionized water at room temperature, with deionized water as the dispersing medium. And the light shading percentage was between 10% and 15% in order to avoid the multiple scattering effects of proteins and to ensure the accuracy of the measurement. The relative refractive indices of the micro-gel particles and the dispersing medium (deionized water) were 1.52 and 1.33, respectively. Each sample with different treatments was measured three times and the average value was taken.

#### 2.3.2. Free Sulfhydryl Determination of PPI and PPMs

Free sulfhydryl (SH_F_) was measured according to Yang et al. [20], with minor modifications. Each sample was diluted to 10 mg/mL and then mixed with 3 mL of Tris-glycine buffer (0.086 M Tris-glycine, 0.004 M EDTA, pH 8.0) and 0.03 mL of Ellman’s reagent (4.0 mg/mL DTNB) in the same buffer. After fully dissolving, the mixtures were centrifuged at 8000 rpm for 10 min. A Tris-glycine buffer and Ellman reagent were added into the supernatant and mixed evenly. A412 was determined by UV spectrophotometer (UV-2500, Shimadzu, Shanghai, China) after 25 min. In the experiment, the solution with the Ellman reagent without protein was used as the control. SH_F_ content was calculated according to Equation (1), written as
(1)$${SH}_{F}(\mu mol/g)=\frac{73.53A_{412}\times D}{C}$$
where *D* is dilution ratio and *C* is the protein concentration (mg/mL).

#### 2.3.3. Fourier Transform Infrared Spectroscopy of PPI and PPMs

The wavenumber range of the iS10 FTIR spectrometer (Thermo Nicolet, Madison, WI, USA) of Nicole Power was 4000–400 cm^−1^. The resolution of the spectrometer was 4 cm^−1^. The signal to noise ratio was 50,000:1, and the scanning count was 32 times [21].

### 2.4. Emulsifying Properties of PPI and PPMs

Emulsifying activity index (EAI) and emulsion stability index (ESI) were measured according to Figueroa-González et al. [22], with minor modifications. A total of 2 mL corn oil and 8 mL protein solution were mixed. Then, the mixture was homogenized by using a T18 Ultra-Turrax (IKA, Staufen, Germany) at 10,000 rpm for 3 min. Immediately, 50 µL of the emulsion was diluted with 2.45 mL 0.1% SDS solution. The A0 and A10 of the diluted emulsion after 0 min and 10 min of resting at 500 nm were recorded. Then, the EAI and ESI values were calculated using Equations (2) and (3), written as
(2)$$\mathrm{EAI}\;(m^{2}/g)=\frac{2\;\times \;2.303\;\times \;A_{0}\;\times \;\mathrm{DF}}{C\;\times \;\varphi \;\times \;10000}$$
(3)$$\mathrm{ESI}\;(\%)=\frac{A_{10}}{A_{0}}\;\times \;100\%$$
where φ is the volume fraction of corn oil (*v*/*v*) (φ = 0.20), DF denotes dilution ratio (200), and C is the concentration of emulsifier (g/mL).

### 2.5. Emulsion Preparation and Characterization

#### 2.5.1. Emulsion Preparation

A 60 wt% oil phase and 40 wt% PPI or PPM (pH 7.0) were mixed to prepare the Pickering emulsion. The mixture was sheared at 10,000 rpm for 5 min by using the T18 Ultra-Turrax (IKA, Staufen, Germany).

#### 2.5.2. Microstructure Observation

The microstructure of the emulsion droplets was observed using LSM800 CLSM (Zeiss, Bartenwueburg, Germany). A total of 20 μL of 0.1% (*w*/*v*) Nile red solution was used as a fluorescent dye of oil in the 1 mL emulsion, with excitation and emission wavelengths at 633 nm and 488 nm, respectively [23].

#### 2.5.3. Rheological Properties

##### Dynamic Oscillatory Measurements

The emulsion was characterized by a MARS 60 rheometer (Haake, Karlsruhe, Ger-many) at 25 °C [24]. The storage modulus *G*′ (Pa), loss modulus *G*″ (Pa), and tan δ of the emulsions, with frequency ranging from 0.01 to 10 Hz, were measured using a clamp with a conical (25 mm) diameter and a distance of 1 mm between parallel plates. *G*′, *G*″, and the angular frequency *ω* were fitted using Equations (4) and (5), written as
(4)$$G^{\prime }=K^{\prime }{\left(\omega \right)}^{n^{\prime }}$$
(5)$$G^{\prime \prime }=K^{\prime \prime }{\left(\omega \right)}^{n^{\prime \prime }}$$
where *K*′ (Pa·s^n′^) and *K*″ (Pa·s^n″^) are constants and *n*′ and *n*″ may be referred to as the frequency exponents.

##### Steady-State Shear Properties

The steady-state shear properties of the PPM (pH 7.0)-stabilized emulsion were measured by the MARS 60 rheometer (Haake, Karlsruhe, Germany) at 25 °C. Shear rate was set to increase from 1 s^−1^ to 300 s^−1^ in 400 s [25]. Shear rate (*γ*, s^−1^) and apparent viscosity (η, Pa·s) were fitted by a power-law model (Equation (6)). Then, the consistency coefficient *K* (Pa·s^n^) and flow behavior index *n* were obtained [10].
(6)$$\eta =K\;\gamma^{n-1}\;$$

#### 2.5.4. In Vitro Digestion

##### In Vitro Digestion Model

The in vitro digestion model, including three digestion stages of saliva, stomach, and intestine, was created [26,27]. The formulations of the stimulated saliva fluid (SSF), stimulated gastric fluid (SGF), and stimulated intestinal fluid (SIF) are shown in Appendix A. 

In general, for the oral phase, 5 g of the emulsions were mixed with 4 mL SSF electrolyte stock solution (contained 3.75 mg/mL mucin). Then, 0.975 mL of deionized water and 25 μL CaCl_2_ solution (0.3 mol/L) were added to the mixture. After the pH was adjusted to 7.0, the mixtures were then stirred at 200 rpm for 2 min (37 °C).

For the gastric phase, 7.5 mL SGF, 1.6 mL porcine pepsin solution (25,000 U/mL), 5 μL of 0.3 mol/L CaCl_2_ solution, and 148 μL of 6 mol/L HCl solution were added into 10 mL of the above mixture while stirring at 300 rpm for 2 h at 37 °C.

For the intestinal phase, after the pH was adjusted to 7.0 using a 1 mol/L NaOH solution, the gastric chyme for each sample was mixed with a 18.5 mL SIF and 40 μL of 0.3 mol/L CaCl_2_ solution. The mixtures were stirred at 200 rpm for 2 h at 37 °C.

##### Particle Size Distribution and Microstructure of Emulsions during In Vitro Digestion

Particle size distribution and microscopic images of PPI-stabilized emulsion and PPM-stabilized Pickering emulsion during in vitro digestion were measured according to the methods in Section 2.3.1 and Section 2.5.2, respectively.

##### Free Fatty Acids (FFA) Release of Emulsions

During simulated intestinal digestion, lipids were continuously hydrolyzed to *FFA* by pancreatic lipase. The amount of FFAs released from the system was calculated based on the volume of the NaOH solution (0.1 mol/L) consumed in 2 h according to Equation (7), written as
(7)$$FFA\%=100\times \frac{V_{NaOH}\times 0.1\times M_{Lipid}}{2\times m_{Lipid}}$$
where *M_Lipid_* is the molar mass of corn oil (872 g/mol) and *m_Lipid_* is the weight of corn oil in the emulsions (g).

### 2.6. Ice Cream Preparation and Characterization

#### 2.6.1. Ice Cream Preparation

Different substitution rates (0%, 20%, 40%, 60%, 80%, and 100%) of cream in the ice cream were substituted by PPM(pH 7.0)-stabilized emulsion (Table 1). And the samples were named PR0, PR20, PR40, PR60, PR80, and PR100. They were homogenized for 5 min at 3000 rpm after mixing and then were pasteurized at 72 °C for 30 min. They were cooled and stored at 4 °C for 4 h. Finally, the finished product was obtained after pouring them into an ice cream maker (Pink Bunny, Zhongshan, China). Three parallel experiments for each sample were carried out.

#### 2.6.2. Rheological Properties

Rheological properties of ice cream slurries were measured according to the method of Section 2.5.3.

#### 2.6.3. Melting Measurement

After extrusion of ice cream, about 30 g of ice cream (the mass was recorded as *M*_0_) was weighed and placed on a metal net with a sieve size of 3 mm × 3 mm at 25 °C [28]. The dropping mass of ice cream was weighed every 10 min until it reached 50 min. Melting and melting rate were calculated according to Equations (8) and (9), written as
(8)$$Melting\;(\%)=\frac{M_{0}}{M_{50}}\times 100\%$$
(9)$$Melting\;rate\;(g/min)=\frac{Mass\;of\;dropping}{Melting\;time}$$
where *M*_50_ is the mass of ice cream after 50 min of dropping.

#### 2.6.4. Overrun Measurement

The overrun of ice cream was measured by recording per volume weight of ice cream slurry (m) and per volume weight of ice cream (m_f_) according to Equation (10) [29].
(10)$$Overrun\left(\%\right)=\frac{m-m_{f}}{m_{f}}\times 100\%$$

#### 2.6.5. Sensory Evaluation

The sensory evaluation methods were based on the methods of Roberta et al. [30], with some modifications. According to the Chinese standard [31] (criterion for sensory evaluation of frozen drinks and ice cream), 20 reviewers were trained for 20 days to form the review panel. Samples were divided as separate portions in 30 mL plastic cups and were assessed in duplicate. The samples were coded, and their order randomized. Ice cream samples were prepared and stored at −18 °C. Before evaluation, the samples were left at room temperature for 5 min. The samples were evaluated by the reviewers in turn for the following five main aspects: color, morphology, organization, taste and odor, and impurity. Distilled water was required to eliminate the effects between samples after different flavors. A 100-point intensity scale (1–10 points for color, 6–30 points for morphology, 5–30 points for organization, 5–20 points for taste and odor, and 3–10 points for impurity) was used for each term, as detailed in Table 2. Sensory acceptance was obtained by adding up the scores of the color, morphology, organization, taste and odor, and impurity. 

### 2.7. Statistical Analysis

All the experiments were repeated at least three times. All data were assed with one-way ANOVA followed with Tukey’s multiple-range test. Differences were considered statistically significant at *p* < 0.05. The results were resented as mean ± standard deviations. 

## 3. Results and Discussion

### 3.1. Protein Structure Analysis

#### 3.1.1. Particle Size Distribution of PPMs

The mean particle sizes of the PPMs prepared at different pH ranging from 4.0 to 7.0 are shown in Figure 1A. It shows that the pH has a significant effect on the particle size distribution of the prepared PPMs. During microparticulation, pea protein was thermally denatured to form a gel structure. After high-speed homogenization, the gel was crushed into a three-dimensional mesh structure with a diameter of 13.65–56.78 μm. The mean particle size of PPM (pH 6.0) was the maximum and was 56.78 μm. A decline of mean particle size was observed when the microparticulation pH was far from pH 6.0. Mean particle sizes of PPM (pH 4.0), PPM (pH 4.5), and PPM (pH 5.0) were 20.69 μm, 18.79 μm, and 20.80 μm, respectively. There was no significant difference among the mean particle sizes of PPM (pH 4.0), PPM (pH 4.5), and PPM (pH 5.0). The sample with the smallest particle size was PPM (pH 7.0), with a mean particle size of 13.65 μm.

Guo et al. [32], Cui et al. [33], and Schmitt et al. [34] confirmed that the surface charge is close to zero at the isoelectric point (pI). Thus, protein molecules aggregate and precipitate under hydrophobic interactions. However, when the pH is far away from the pI, the surface charge of protein molecules increases, which generates intermolecular repulsion and inhibits aggregation, leading to a decrease in protein particle size.

#### 3.1.2. Content of Free Sulfhydryl Groups

Free sulfhydryl groups (SH_F_) reflect the changes of the protein tertiary structure and played key role in the protein rigid structure formation. Free sulfhydryl groups can form disulfide bonds after oxidation, and disulfide bonds can be further reduced to free sulfhydryl groups through hydrogenation under the conditions of mutual conversion during heat treatment [35] and shear treatment.

The free sulfhydryl group content of different PPMs is shown in Figure 1B. SH_F_ content of PPI is 5.45 μmol/g. The SH_F_ content of PPM (pH 4.0), PPM (pH 4.5), PPM (pH 5.0), PPM (pH 5.5), PPM (pH 6.0), and PPM (pH 6.5) were 0.07, 0.09, 0.11, 0.19, 0.64, and 4.75 μmol/g, which were significantly lower than PPI; it was due to the highly concentrated, accessible, and reactive free sulfhydryl groups that were oxidized to form the disulfide bond [36]. 

The SH_F_ content of PPMs gradually increased with the increasing microparticulation pH. The SH_F_ content of PPM (pH 7.0) was the highest (11.10 μmol/g) and significantly higher than PPI. The increasing content of the free sulfhydryl groups may be due to the unfolding of the protein molecular structure and the break of disulfide bonds caused by high-temperature heating. Mession et al. [37] confirmed that heating causes the unfolding of protein molecules and exposing of embedded free sulfhydryl groups, disulfide bonds, and hydrophobic groups. The change of the functional group distribution can enhance the intermolecular interaction and then promote the formation of the gel network [38].

Guo et al. [10] found that the SH_F_ content of protein molecules increases the promotion of the interaction between protein molecules and forms a tight and uniform network of gels. In summary, pH and heat can change the content of sulfhydryl bonds and affect the functional properties of proteins. 

#### 3.1.3. Fourier Transform Infrared Spectroscopy Analysis of PPI and PPMs

Figure 1C displays the spectra of PPI and PPMs in the 4000 to 400 cm^−1^ range. The OMNIC software was used to fit the peaks in the range of 1700 to 1600 cm^−1^ to calculate the content of each secondary structure of proteins [39]. The peaks located at 1650–1658 cm^−1^ reflect the α-helix, 1610–1640 cm^−1^ reflect the β-sheet, 1660–1700 cm^−1^ reflect the β-turn, and the peaks at 1640–1650 cm^−1^ reflect the random coil [40]; the results are shown in Table 3. As can be seen in Table 3, the PPI contains 11.66% α-helix, 33.04% β-sheet, 46.94% β-turn, and 8.36% random coil. The β-sheet and β-turn are the dominant secondary structures of PPI and PPMs, followed by random coil and α-helix. Compared with PPI, the proportion of α-helix significantly decreased, which is consistent with the research of Beck et al. [41]. The β-sheet content in PPM (pH 4.0) and PPM (pH 4.5) significantly increased to 41.64% and 39.73%, respectively; the β-turn content had significantly decreased to 39.53% and 41.76%, respectively; random coil content showed no significant difference with PPI. The decreasing α-helix contents and increasing random coil content and β-sheet content were observed in PPM (pH 5.0) and PPM (pH 6.0). The secondary structures of PPM (pH 5.5) and PPM (pH 6.5) shifted from β-sheet and α-helix to β-turn and random coil, which is consistent with the results of Sun et al. [16]. It can be concluded that the reduction in α-helical structures might have been due to the fact that microparticulation destroyed the rigid structure of protein molecules, causing unfolding and rearrangement. The rearranged protein molecules tend to have a higher degree of intermolecular flexibility [39]. There is no significant difference in the content of each secondary structure between PPM (pH 7.0) and PPI. Therefore, it can be concluded that the different pH during macroparticulation influenced the degree of protein unfolding, the distribution of amino acid side chains, and changes in protein surface charges and [42]. Thus, flexible structures in the protein micro-gel particle formed after the disruption of the natural structures during microparticulation could lead to differences in their physiochemical properties with PPI, such as interfacial properties [16].

### 3.2. Emulsifying Properties

Figure 2A shows that the EAI of PPI increased gradually with the increase of pH from 5.5 to 7.0. Different pH preparation of PPMs also had a significant effect on EAI (Figure 2A). The EAI at pH near the pI (pH 4.0–6.0) was lower than the EAI at pH 6.0–7.0. The EAI values of the protein micro-gel particles prepared at pH 6.0–7.0 were higher than that of pI (pH 4.0–6.0), which was mainly due to the higher degree of protein molecules unfolding. The unfolded protein molecules could rapidly adsorb on the oil–water interface to form a denser viscoelastic interfacial layer, which stabilized the dispersed oil droplets. At neutral pH far from the isoelectric point, pea protein forms a stronger and denser viscoelastic network when adsorbed at the oil–water interface than that at pI (pH 4.0–6.0) [43]. Near the isoelectric point, protein molecules have no surface charge, so low intermolecular repulsion leads to molecular aggregation, which caused the lower EAI of proteins [44]. Another reason could be the proper three-phase contact angle of the micro-gel particle, which is attributed to balanced distribution of hydrophilic and hydrophobic groups of PPM prepared at pH 6.0–7.0 [33].

The pH values of the dispersions have significant impacts on the ESI of PPI (Figure 2B). Among them, PPI has the better ESI, at pH 4.0 and pH 7.0, than others, with values of 89.10% and 92.40%, respectively. Except for PPM (pH 4.0), different preparation pH had no significant effect on the ESI values of other PPM samples. It can be seen that protein micro-gel particles once adsorbed on the oil–water interface, form a stable interfacial layer to stabilize the oil droplets [45].

The EAI and ESI of PPMs were both significantly higher than that of the PPI at corresponding pH (Figure 2), indicating that micro-gel particles have higher potential applications in multiphase systems than PPI. Natural pea proteins are globular proteins with well-defined molecular conformations, where hydrophilic groups are distributed on the surface of the protein molecule, while the hydrophobic groups are embedded within the molecule. During microparticulation, protein molecules partially unfolded, then recombined and cross-linked to form the gel network structure [36]. Subsequently, high-speed shear treatment disrupts the gel maintained by non-covalent bonds to obtain micro-gel particles. Simultaneously, the surface hydrophilic and hydrophobic groups of protein molecules undergo rearrangement [45]. In summary, compared to the rigid structure of PPI, more flexible structures were formed during microparticulation, which can be better adsorbed on the oil–water interface to form an interfacial layer that stabilizes the oil droplets. The higher EAI (56.56 m^2^/g) and ESI (95.10%) of PPM (pH 7.0) than other samples (Figure 2) indicated that PPM (pH 7.0) is more suitable for application in O/W emulsion stabilization. Therefore, PPM (pH 7.0) was chosen for subsequent emulsion preparation and research.

### 3.3. Characterization of the PPM (pH 7.0)-Stabilized Pickering Emulsion

#### 3.3.1. Micromorphology Observation

After staining with Nile red, the oil droplets showed red after excitation at 636 nm. It can be seen from Figure 3A that oil droplets exist in the emulsion in a round shape with uniform size. There is an interfacial protein layer, which can not only effectively reduce the surface tension, but also help to form uniform and stable emulsion [46]. This also explains well the better ESI of PPM (pH 7.0). And it can be found that the droplets size of the emulsion was less than 10 μm, which indicated that the PPM (pH 7.0)-stabilized emulsion can simulate the smooth and delicate texture of saturated fat and can be used as a saturated fat replacement to replace thin cream to prepare low-saturated fat ice cream.

#### 3.3.2. Rheological Properties

Dynamic rheological measurements can reflect the formation and destruction of composite gel matrix [24]. The viscoelasticity of the PPM (pH 7.0)-stabilized Pickering emulsion can be characterized by storage modulus (G′) and loss modulus (G″). Figure 3B shows that G′ and G″ of the PPM (pH 7.0)-stabilized Pickering emulsion slightly increased with the increasing frequency. The slopes of G′ (K′ = 4753.64 Pa·s^n′^) were higher than that of G″ (K″ = 768.06 Pa·s^n″^), which indicated that G′ was more frequency dependent than G″ [10]. G′ of the PPM (pH 7.0)-stabilized Pickering emulsion was always larger than G″ and tan δ was less than one throughout the tested frequency range, indicating that elastic behavior dominated the rheological properties [47]. 

Figure 3C shows that the viscosity of the PPM (pH 7.0)-stabilized Pickering emulsion decreased with the increase in shear rate, indicating that the emulsion is a shear-thinning fluid. In PPM (pH 7.0)-stabilized Pickering emulsion, the droplets are close enough together to interact with each other which may lead to the formation of a three-dimensional network of aggregated droplets. As the shear rate is increased, the hydrodynamic forces cause aggregates to become deformed and eventually disrupted which results in a reduction in the viscosity. The flow behavior index *n* is 0.81, which is less than 1. It indicated that the PPM (pH 7.0)-stabilized Pickering emulsion is pseudoplastic fluid. According to Long et al. [48], cream is a pseudoplastic fluid and exhibits shear-thinning behavior. PPM (pH 7.0)-stabilized Pickering emulsion showed the same rheology properties as cream, which ensured its potential application as a saturated fat replacement for cream.

#### 3.3.3. In Vitro Digestion

Figure 4A,B shows the particle size distribution and microstructure of PPI-stabilized emulsion and PPM (pH 7.0)-stabilized Pickering emulsion during simulated in vitro digestion. Figure 4A_1_,B_1_ shows that before digestion, the oil droplets in both emulsions exhibited uniformly distribution in the form of round balls. The particle sizes d_32_ in the two emulsions are not significantly different. However, there is a very small distribution peak at 10–100 μm in the particle size distribution curve of the PPI-stabilized emulsion, indicating the presence of a few large oil droplets, which leads to a significantly higher d_43_ (8.52 μm) than that of the PPM (pH 7.0)-stabilized Pickering emulsion (4.45 μm). Comparing Figure 4A_2_,B_2_ and Figure 4A_1_,B_1_, it can be found that after a short period of oral digestion, the particle size distribution and microscopic morphology of the two emulsions did not change significantly (*p* > 0.05). This can also be confirmed by the non-significantly different values of d_32_ and d_43_.

After simulated gastric digestion, the particle size distribution and microscopic morphology of both emulsions changed dramatically (Figure 4A_3_,B_3_). The oil droplet size increased significantly (*p* < 0.05), which was attributed to the coalescence of the oil droplets. The gastric acid (pH 2.0) and pepsin in the SGF caused the protein at the oil–water interface in the emulsions to be denatured, losing the role of forming interfacial film to protect the oil droplets. It further led to flocculation and even coalescence of the oil droplets [49]. Compared to PPM (pH 7.0)-stabilized Pickering emulsions, PPI-stabilized emulsions showed more flocculation of oil droplets after simulated gastric digestion (Figure 4A_3_,B_3_). This suggests that the micro-gel particles formed by pea proteins have an enhanced ability to resist both the acidic environment and pepsin digestion [50]. 

After simulated intestinal digestion, the number of oil drops in the two emulsions decreased significantly under the dilution of digestive fluid and the digestion of lipase-colipase (Figure 4A_4_,B_4_); and the oil droplet size significantly increased (*p* < 0.05). However, the oil droplet size in PPM (pH 7.0)-stabilized Pickering emulsion was significantly smaller than that in PPI-stabilized emulsion (*p* < 0.05). And the number of oil droplets was significantly greater than that in PPI stabilized emulsion (*p* < 0.05). It indicated that PPM (pH 7.0) has the ability to protect the integrity of the emulsion structure in the simulated gastrointestinal fluid and resist the digestion of simulated gastrointestinal fluid. This can also be confirmed from Figure 4C; it shows that the FFA release rate of both emulsions increases rapidly within 45 min of simulated intestinal digestion. After 45 min, the FFA release rate slowed down and eventually approached 0. Obviously, the FFA release rate of PPM (pH 7.0)-stabilized Pickering emulsion was significantly less than that of the PPI-stabilized emulsion (*p* < 0.05). This indicates that the PPM (pH 7.0) on the oil–water interface has the ability to block the adsorption of bile salts and lipase, thus the delay of the lipid digestion of emulsions. The results of Sarkar et al. [51] also highlighted that the protein gel is able to slow the diffusion of both lipase–colipase and proteases, thus inhibiting lipid digestion of O/W emulsion. Therefore, PPM (pH 7.0)-stabilized emulsion can be used as compound fat replacement to reduce the lipid content of food and decline the lipid bioavailability.

### 3.4. Steady-State Shear Properties of Ice Cream Slurry

As can be seen from Figure 5, the viscosity of all ice cream slurries show a decreasing trend with the increasing shear rate and show a nonlinear relationship, indicating that all ice cream slurries exhibited shear-thinning behavior. This is due to the fact that at rest or low flow rates, the macromolecular chain-like structures are entangled with each other, and the viscosity of all ice cream slurries is high. With the increasing shear rate, due to the effect of the shear stress, the disordered macromolecular chain rolled and rotated and contracted into clusters, which reduced the intermolecular cross-linking, and thus the viscosity gradually decreased [52].

According to the power law model with high regression coefficients (R^2^ = 0.98–0.99), the fitting parameters were used to obtain the flow behavior index (*n*) and consistency coefficient (K) of all ice cream slurries. As can be seen in Figure 5, there were no significant differences in the n values of the ice cream slurries as the substitution rate increased from 0% to 100%. The index n is greater than zero and less than one, indicating that all ice cream slurries are pseudoplastic fluids. K values significantly increased with the increasing substitution rate, which indicated that the ice cream slurries showed more viscous properties due to the increase in apparent viscosity. Wang et al. [53] also found that the K values of ice cream slurries increased with the increase of soybean oil body substitution rate. They claimed that it was due to the fact that the increased content of Pickering emulsion introduces PPM biomolecules, whose hydrogen bonding or hydrophobic interactions between the nonpolar side segments of the carbon skeleton produce new ordered and improved structures, which lead to differences in rheological behavior.

### 3.5. Dynamic Oscillatory Measurements of Ice Cream Slurry

The dynamic oscillatory of ice cream slurries can reflect the physicochemical properties and flow behavior during chewing in the mouth. Storage modulus G′ and loss modulus G″ were analyzed, shown in Figure 6A,B. G′ and G″ of all ice cream slurries, except PR40 and PR60, increased with increasing frequency until 1.3 Hz, which indicated the destruction of the structures. G′ and G″ of PR40 and PR60 remained stable until 2.8 Hz, indicating that the structure was still intact. This also indicates the enhanced structural stability of low-fat ice cream slurries at 40% and 60% substitution rates. The tan δ (G″/G′) of ice cream slurries is a key indicator of the phase melting and structural changes [54], which are shown in Figure 6C. The tan δ of all the samples in the measured frequency range are less than one, indicating that the G′ values are higher than the G″ values, which suggests that all the ice cream slurries dominate the liquid characteristics, which are also known as viscoelastic fluids; viscoelastic fluid characteristics are the desired rheological properties of ice cream. It can be seen that the application of PPM (pH 7.0)-stabilized Pickering emulsion replacement of saturated fat in ice cream did not change its solid/liquid behavior. This also indicates PPM (pH 7.0)-stabilized Pickering emulsion has the potential to be applied in low-fat foods as compound fat replacements.

### 3.6. Physicochemical Properties and Sensory Evaluation of Ice Cream

Table 4 shows the effects of the substitution rates of the PPM (pH 7.0)-stabilized Pickering emulsion on the physicochemical properties and sensory evaluation of the ice cream. There was no significant difference in the melting of each ice cream as the substitution rate increased to 80%. This indicates that the substitution of saturated fat by a moderate amount of PPM (pH 7.0)-stabilized Pickering emulsion does not affect the ice phase organization. However, the melting of PR100 (1.345%) was significantly greater than the other samples, due to the fact that the ice phase arrangement of PR100 was different from the other samples. In the PR100 with a lower overrun and stability, there was not enough network space structure to prevent melting and complete collapse. According to Kaleda et al. [55], the difference in melting between the two types of ice cream can only be attributed to differences in ice phase arrangement.

The melting rate results showed that there was no significant difference among the melting rates of PR20, PR40, and PR0. However, as the substitution rate continued to increase to 80%, the melting rate significantly increased to 0.46 g/min. This may be related to the fact that protein micro-gel particles in the emulsion increase the protein content of the ice cream, which alters the melting rate of the final product [56]. When the PPM (pH 7.0)-stabilized Pickering emulsion completely replaced the saturated fat in the ice cream, the melting rate was 0.33 g/min rather lower than PR80.

With an increase in the substitution ratio, the overrun of PR20 and PR40 increased from 85.70% of PR0 to 131.90% of PR20 and 103.70% of PR40, respectively. The overrun of PR20 and PR40, respectively, was 131.90% and 103.70%, which was significantly higher than the other samples. The higher overrun of PR20 and PR40 was related to the lower viscosity than other samples (Figure 5). The overrun was positively correlated with the amount of air introduced during the ice cream production [57]. The low viscosity of PR20 and PR40 facilitated a large amount of air during vigorous mixing. The ice cream with suitable overrun (90–110%) had a soft and delicate taste, a strong melting resistance, an unstable shape, and a difficult deformation [58]. The overrun of PR40, PR60, and PR100 were all within this range, which may result in higher sensory evaluation than PR20 and PR80 (Table 4).

The presence of a slight beany smell was the main reason why the sensory evaluation of PR100 was significantly lower than that of PR40 and PR60. The obvious ice crystal taste was the main reason for the lowest sensory score of PR80. The PR0 had a higher sensory evaluation and no significant difference compared to PR40 and PR60, which was due to its higher saturated fat content. Mahdian and Karazhian [59] confirmed that the sensory evaluation of ice cream was related to the content of cream. Reducing saturated fat of ice cream could result in structural defects, such as ice crystals and a rough texture.

## 4. Conclusions

PPM (pH 7.0) was suitable for application in O/W emulsion stabilization because of its proper particle size, more flexible structure, and high EAI and ESI. The Pickering emulsion stabilized by PPM (pH 7.0) had a uniform oil droplet distribution and similar rheological properties to cream; therefore, it can be used as a saturated fat replacement in the production of ice cream. PPM (pH 7.0)-stabilized emulsion can be used to substitute 60% cream to produce low-saturated fat ice cream, which had high structural stability, similar melting properties, overrun, and sensory properties with PR0. The results indicated that PPM (pH 7.0)-stabilized Pickering emulsion has a similar fatty acid profile to that of the corn oil used, but the rheological properties similar to that of saturated fatty acid. Therefore, it is feasible to use it as a saturated fat replacement in the production of low-saturated fat food to meet the demand of consumers for unsaturated fat.

## Figures and Tables

**Figure 1 foods-13-01511-f001:**
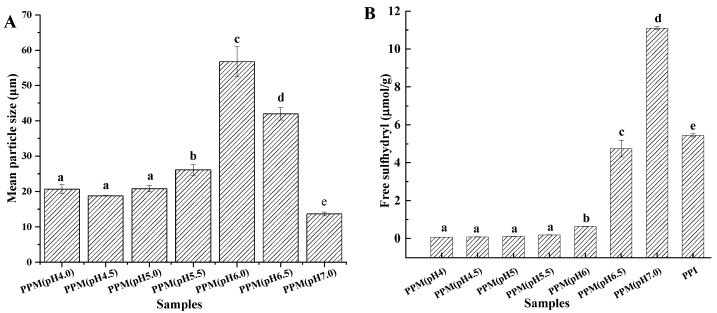
Particle size distribution (**A**), free sulfhydryl content (**B**) and FTIR spectrum (**C**) of pea protein micro−gels (PPMs) and pea protein isolate (PPI). The values that do not bear the same letter on the column are significantly different (*p* < 0.05).

**Figure 2 foods-13-01511-f002:**
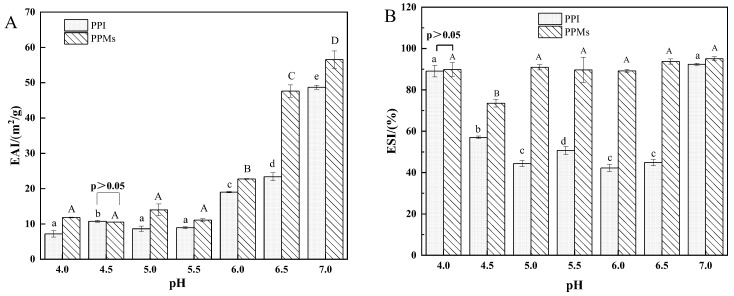
Emulsifying activity index (**A**) and emulsion stability index (**B**) of PPI at different pH and PPMs. The values that do not bear the same letter on the column are significantly different (*p* < 0.05).

**Figure 3 foods-13-01511-f003:**
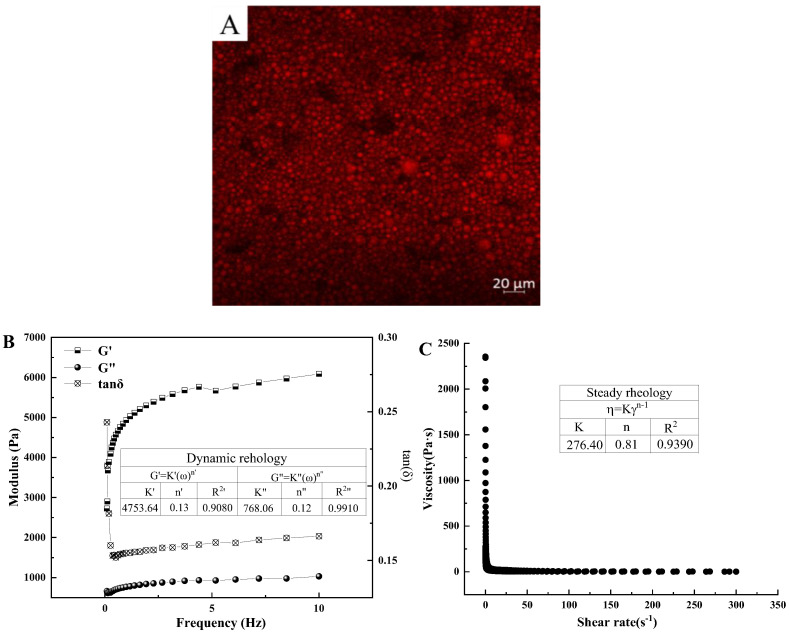
Microstructure (**A**), dynamic mechanical spectra, (**B**) and steady flow curves (**C**) of the PPM (pH 7.0)-stabilized Pickering emulsion.

**Figure 4 foods-13-01511-f004:**
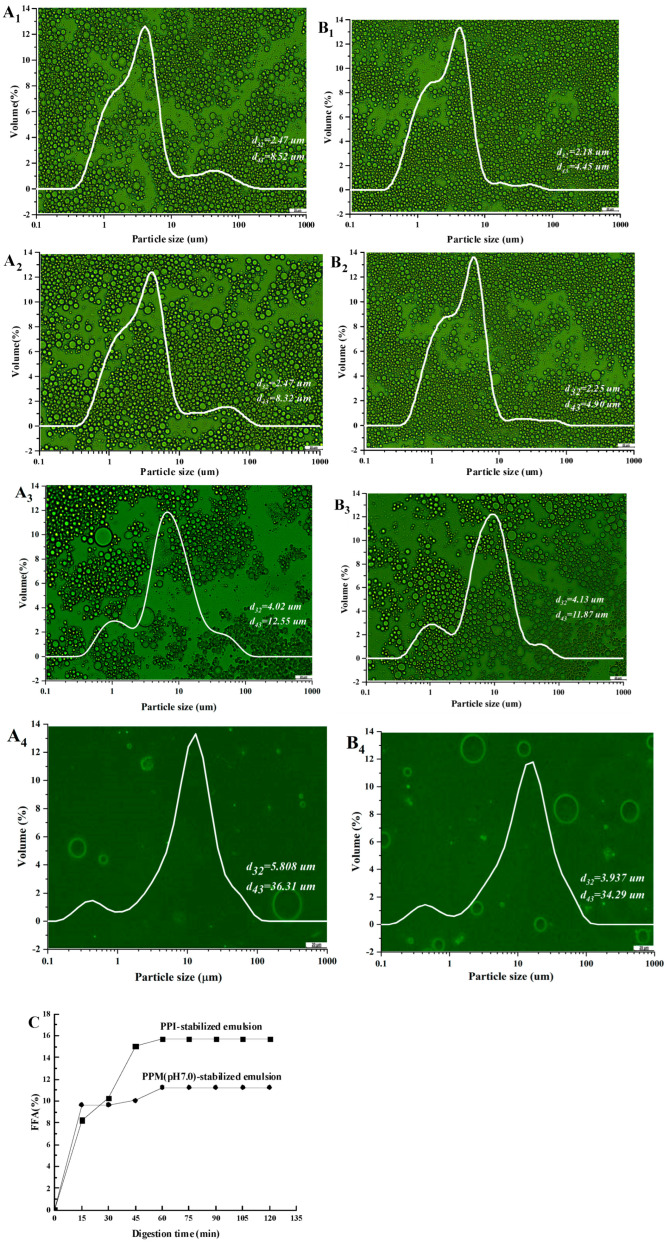
The microstructure, particle size distribution, and free fatty acids (FFA) release (**C**) of the PPI−stabilized emulsion (**A**) and PPM (pH 7.0)-stabilized Pickering emulsion (**B**) during in vitro digestion. (**A_1_**,**B_1_**) Initial, (**A_2_**,**B_2_**) Oral phase, (**A_3_**,**B_3_**) Gastric phase, (**A_4_**,**B_4_**) Intestinal phase.

**Figure 5 foods-13-01511-f005:**
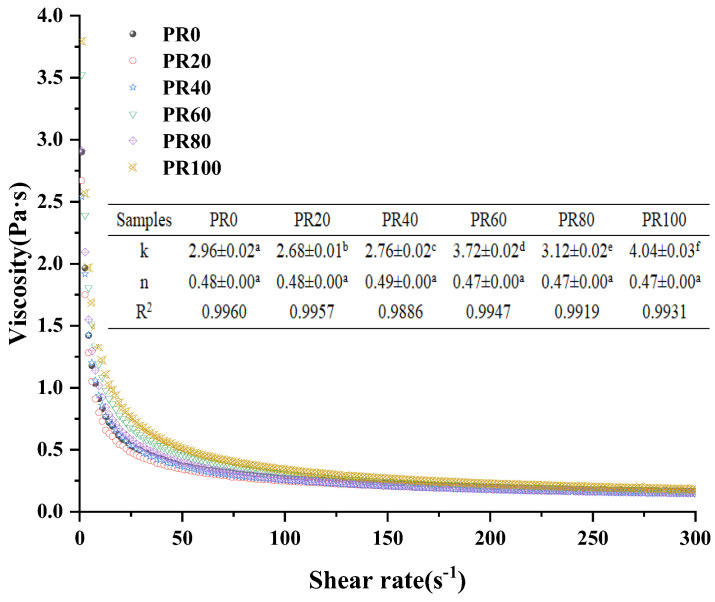
Flow curves of ice cream slurries measured at 25 °C. ^a−f^: The values that do not bear the same letter on the column are significantly different (*p* < 0.05).

**Figure 6 foods-13-01511-f006:**
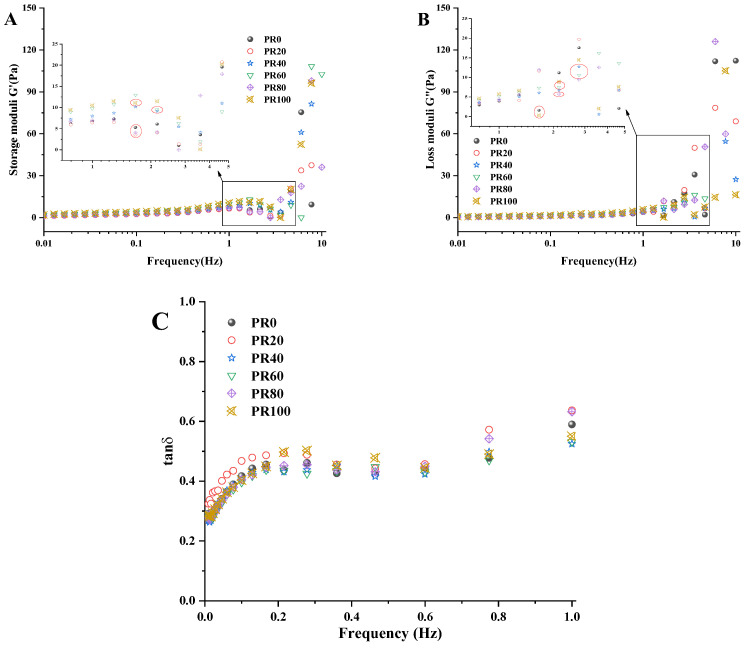
Dynamic mechanical spectra ((**A**) storage modulus, (**B**) loss modulus, (**C**) tan δ) of ice cream slurries.

**Table 1 foods-13-01511-t001:** Composition of ice cream at different saturated fat replacer ratios.

Material	Samples					
	PR0	PR20	PR40	PR60	PR80	PR100
Water/mL	225	225	225	225	225	225
Skim milk/g	36	36	36	36	36	36
Cream/g	27	21.6	16.2	10.8	5.4	0
Emulsion/g	0	5.4	10.8	16.2	21.6	27
Sucralose/g	0.04	0.04	0.04	0.04	0.04	0.04
Sodium alginate/g	0.9	0.9	0.9	0.9	0.9	0.9

**Table 2 foods-13-01511-t002:** Sensory evaluation scoring index and criteria of ice cream.

Item	Sensory Evaluation Criteria	Scoring Criteria
Color	White or light yellow and uniform.	9~10
	Color is too light or too dark.	5~8
	There are different colors.	1~4
Morphology	Complete shape, uniform volume, no distortion.	27~30
	Incomplete shape, general uniform volume, slightly twisted and deformed.	20~26
	Appearance of clots or too sticky and other undesirable phenomena.	11~19
	Distorted appearance, incomplete, uneven.	6~10
Organization	Soft and dense, no air holes appear.	28~30
	Ice crystals are found in the tissue.	21~27
	Serious ice crystals are found in the tissue.	14~20
	The tissue structure is not fluffy enough or too fluffy.	5~13
Taste and odor	Soft milky flavor, no bean smell.	17~20
	Insufficient light milky flavor, with soya odor or too sweet.	13~16
	Bean-like flavor is serious or other foreign odor appears.	5~12
Impurity	No visible impurities.	7~10
	There are visible impurities.	3~6

**Table 3 foods-13-01511-t003:** Content of secondary structures of PPI and PPMs.

Samples	α-Helix	β-Sheet	β-Turn	Random Coil
PPI	11.66 ± 0.20 ^a^	33.04 ± 0.30 ^cg^	46.94 ± 0.31 ^c^	8.36 ± 0.20 ^b^
PPM (pH 4.0)	10.06 ± 0.26 ^b^	41.64 ± 0.55 ^a^	39.53 ± 1.49 ^e^	8.77 ± 0.67 ^b^
PPM (pH 4.5)	9.84 ± 0.63 ^b^	39.73 ± 0.09 ^b^	41.76 ± 1.44 ^d^	9.04 ± 0.21 ^b^
PPM (pH 5.0)	9.19 ± 0.40 ^b^	32.29 ± 0.33 ^c^	48.35 ± 0.20 ^b^	10.16 ± 0.13 ^a^
PPM (pH 5.5)	10.66 ± 0.76 ^b^	28.57 ± 0.96 ^d^	51.83 ± 0.40 ^a^	8.94 ± 0.60 ^b^
PPM (pH 6.0)	9.63 ± 0.07 ^b^	35.47 ± 1.24 ^e^	46.01 ± 1.50 ^c^	8.89 ± 0.19 ^b^
PPM (pH 6.5)	8.78 ± 0.34 ^c^	30.40 ± 0.09 ^f^	50.01 ± 0.13 ^a^	10.81 ± 0.56 ^a^
PPM (pH 7.0)	10.93 ± 0.11 ^ab^	32.79 ± 0.15 ^cg^	47.63 ± 0.24 ^bc^	8.66 ± 0.20 ^b^

The values that do not bear the same letter in the same column are significantly different (*p* < 0.05).

**Table 4 foods-13-01511-t004:** Effects of fat substitution ratio on physicochemical indexes and sensory evaluation of ice cream.

Samples	Melting (%)	Melting Rate (g/min)	Overrun (%)	Sensory Evaluation
PR0	0.64 ± 0.02 ^a^	0.23 ± 0.00 ^a^	85.70 ± 0.71 ^a^	93.7 ± 1.2 ^a^
PR20	0.63 ± 0.02 ^a^	0.24 ± 0.02 ^a^	131.90 ± 3.25 ^b^	89.7 ± 0.6 ^b^
PR40	0.78 ± 0.05 ^a^	0.25 ± 0.01 ^a^	103.70 ± 9.62 ^c^	92.0 ± 1.0 ^ac^
PR60	0.50 ± 0.02 ^a^	0.31 ± 0.01 ^b^	93.13 ± 1.41 ^acd^	95.0 ± 0 ^a^
PR80	0.40 ± 0.01 ^a^	0.46 ± 0.04 ^c^	82.70 ± 2.12 ^ade^	83.0 ± 1.0 ^e^
PR100	1.35 ± 0.40 ^f^	0.33 ± 0.02 ^b^	94.75 ± 2.90 ^acd^	91.8 ± 1.5 ^c^

The values do not bear the same letter in the same column are significantly different (*p* < 0.05).

## Data Availability

The original contributions presented in the study are included in the article/Appendix A, further inquiries can be directed to the corresponding author.

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
