# Peer review of "Fabricating Pea Protein Micro-Gel-Stabilized Pickering Emulsion as Saturated Fat Replacement in Ice Cream"

_foods, 2024, doi:10.3390/foods13101511_

Round 1
Reviewer 1 Report
Comments and Suggestions for Authors
General Comments
-Overall, it is an interesting manuscript with a well-defined application for the proposed system. I suggest some changes to improve the manuscript and for it to make a stronger point.
-Grammar should be checked throughout the manuscript as well as the use of plural or singular for words (e.g. line 44 foods and not food, line 49 requires and not require, line 90 microparticulated)
-I think the introduction as well as the abstract should not focus so much on cardiovascular disease for the reason behind this work but on the overall health and well-being of the population. Cardiovascular disease is overly specific, and the introduction does not compose a convincing and proof-based argument to support the claim.
-How do you know that the process for preparing the ice cream (pasteurizing and then cooling) does not affect the microgel structure?
-The methods are not always very well described/explained.
Abstract
-Line 19: it should be “was suitable for application in...”
-Line 28: shows instead of showed.
Introduction
-Line 36: It would be better to use the word humans instead of human beings.
-Line 50: Use researchers or scientists instead of food researchers.
-Line 50: properties instead of qualities.
-Lines 58-61: Some examples of those substitutes you refer to should be mentioned.
-Line 67-69: I believe your last comment should be removed. Yes, the novelty of the substitutes should be highlighted, but the claim that this alone will reduce cardiovascular disease among the population is unbased.
-Lines 73-78: The paragraph needs rephrasing; it is not clear at all. “realize” means understand and its use is not appropriate here. Explain how the structure modification of the proteins will help. Will it change the surface activity properties of the proteins, reduce the droplet size…?
-Lines 88-99: Future tense should be replaced by past tense.
Materials and Methods
-Paragraph 2.3.2: The buffer’s concentration as well as the reagent’s and buffer’s added amounts should be mentioned.
-Paragraph 2.6.5: This is not an acceptable method’s description. How was the test conducted? What method was used for the scoring? Which aspects of the sensory profile were evaluated and how? How was the total score calculated?
-Line 137: were instead of was.
Results
-Paragraph 3.3.1: It is important to state PPI’s theoretical isoelectric point value since you mention it and highlight its importance.
-Lines 275-276: Please rephrase, it is not at all clear.
-Line 287: Micro-, not macro-
-Paragraph 3.1.3: It lacks explanation concerning the changes that were observed. For example, what does the increase of the β-sheet means for the protein? The results should be analyzed more extensively.
-Paragraph 3.3.1: It could be useful to also stain the PPMs and observe the interphase through CLSM and compare the images to the emulsifying index.
-Paragraph 3.3.2: The results should be further analyzed and explain what they offer in understanding the new systems.
-Line 475: “…which be owing to the different ice phase…” this is not a correct grammar type. Change it accordingly.
-Lines 475-477: The term “ice phase organization” should be explained and defined.
-Lines 478-484: The results presented here contradict each other. How is it possible that while with the increase of substitution the melting also increases but that does not happen at 100% substitution?
-Lines 495-501: Could you propose a way to avoid those defects?
Conclusions
-Conclusions should highlight more the major findings of this study and advocate for its novelty and possible applications in the food industry in general. Also, it should be grammar proofed.
Comments on the Quality of English LanguageModerate changes
Author Response
Reviewer 1:
Comment: Overall, it is an interesting manuscript with a well-defined application for the proposed system. I suggest some changes to improve the manuscript and for it to make a stronger point.
Answers: We appreciate the selfless contributions to this manuscript from you. It is because of your insightful and helpful comments and suggestions, we gained the confidence to improve our work better. The manuscript foods-2985301 entitled “Fabricating pea protein microgel-stabilized Pickering emulsion as saturated fat substitutes in ice cream” has been carefully revised with the revisions marked in red. The point-by-point answers to these comments and suggestions were listed as below.
Comment 1: Grammar should be checked throughout the manuscript as well as the use of plural or singular for words (e.g. line 44 foods and not food, line 49 requires and not require, line 90 microparticulated)
Answers: Many thanks for your comment. We have revised the grammar of the manuscript and the plural or singular of words. We have revised "food" into "foods" in line 45, "require" into "requires" in lines line 51, and "microparticulate" into " microparticulated " in line 93 in the revised manuscript.
Comment 2: I think the introduction as well as the abstract should not focus so much on cardiovascular disease for the reason behind this work but on the overall health and well-being of the population. Cardiovascular disease is overly specific, and the introduction does not compose a convincing and proof-based argument to support the claim.
Answers: Thank you for your comments. We have revised the abstract and introduction in lines 13-14 and 37-40 in the revised manuscript which were shown as follows: “Unsaturated fat replacement should be used to reduce the use of saturated fat and trans fatty acids in the diet.”
“Chronic diseases, such as cardiovascular disease, have become the most serious diseases threatening humans in the world today, seriously endangering people's physical health. [1]. Many studies have shown that intake of saturated fats and trans fatty acids increases the risk of chronic disease, whereas intake of unsaturated fats decreases it [2,3].”
Comment 3: How do you know that the process for preparing the ice cream (pasteurizing and then cooling) does not affect the microgel structure?
Answers: Many thanks for the referee’s comment. Whether the structure of the microgel particles was affected by the process or not, the results of the corresponding measurements of the full-saturated and low-satureated fat ice cream in section 2.6.3, 2.6.4 and 2.6.5 verified the possibility of replacing saturated fats with microgel particle-stabilized Pickering emulsions for the preparation of low- saturated fat ice cream.
Comment 4: The methods are not always very well described/explained.
Answers: Thank you for your comments. We have revised the description of the methods in lines 171, 183, section 2.6.1 and section 2.6.5. Please see the revised manuscript.
Comment 5: Abstract-Line 19: it should be “was suitable for application in...”
Answers: Thank you for your comments. We have revised “was suitable application in” into “was suitable for application in” in line 19 in the revised manuscript.
Comment 6: -Line 28: shows instead of showed.
Answers: Thank you for your comments. We have revised “showed” into “shows” in line 29 in the revised manuscript.
Comment 7: Line 36: It would be better to use the word humans instead of human beings.
Answers: Thanks for your comments. We have revised the word humans instead of human beings in line 37 in the revised manuscript.
Comment 8: -Line 50: Use researchers or scientists instead of food researchers.
Answers: Thank you for your comments. We have revised “food researchers” into “researchers or scientists” in line 51 in the revised manuscript.
Comment 9: -Line 50: properties instead of qualities.
Answers: Many thanks for your comment. We have revised “qualities” into “properties” in line 53 in the revised manuscript.
Comment 10: Lines 58-61: Some examples of those substitutes you refer to should be mentioned.
Answers: Many thanks for your comment. We have added some examples of the types of fat substitutes available in lines 58-62 in the revised manuscript which was shown as follows: “Currently, the main approach to reduce saturated fat in food is to use saturated fat substitutes (saturated fat mimics and saturated fat replacers like protein-based fat replacers, carbohydrate-based fat replacers, lipid-based fat replacers and complex fat replacers) to substitute some or all of the saturated fat to ensure sensory properties while reducing the calories.”
Comment 11: Line 67-69: I believe your last comment should be removed. Yes, the novelty of the substitutes should be highlighted, but the claim that this alone will reduce cardiovascular disease among the population is unbased.
Answers: Many thanks for your comment. We have removed the last comment and revised the claims related to cardiovascular disease in lines 68-70 in the revised manuscript which was shown as follows: “Therefore, the development of a novel saturated fat substitute is of positive significance for the development of nutritious and healthy foods.”
Comment 12: Lines 73-78: The paragraph needs rephrasing; it is not clear at all. “realize” means understand and its use is not appropriate here. Explain how the structure modification of the proteins will help. Will it change the surface activity properties of the proteins, reduce the droplet size…?
Answers: Many thanks for your comment. We have revised the paragraph in lines 74-77 in the revised manuscript as follows: “On the one hand, protein-stabilized emulsions enable the solidification of vegetable oils.” And we have revised the sentence in lines 83-87 in the revised manuscript as follows: “Protein micro-gel particles tend to have great interfacial properties because the increasing hydrophobicity of the protein surface during microparticulation. Pickering emulsion stabilized by protein micro-gel particles have advantages such as good stability and biocompatibility.”
Comment 13: Lines 88-99: Future tense should be replaced by past tense.
Answers: Many thanks for your comment. We have revised the future tense of this paragraph to the past tense. The modified sentence was shown in lines 92-100 in the revised manuscript and shown as follows: “Thus, in this study, Pea protein was microparticulated to improve its processing performance, such as emulsification. Different structural and physicochemical properties of pea protein micro-gel particles were obtained by adjusting the preparation pH during microparticulation. Then, an O/W emulsion stabilized by pea protein micro-gel particles was designed to substitute saturated fat and was used as saturated fat replacement in low-saturated fat ice cream. And, the saturated fat in ice cream was partially replaced at different rate of 20%, 40%, 60%, 80% and 100%. The physicochemical properties and rheological properties of full-saturated fat and low-saturated fat ice cream were investigated to find the most suitable replacement ratio.”
Comment 14: Paragraph 2.3.2: The buffer’s concentration as well as the reagent’s and buffer’s added amounts should be mentioned.
Answers: Many thanks for your comment. We have added information on buffer concentrations, reagents and buffer additions. The added information was shown in lines 133-135 in the revised manuscript and shown as follows: “Each sample was diluted to 10 mg/mL and then mixed with 3 mL of Tris-glycine buffer (0.086 M Tris-glycine, 0.004 M EDTA, pH 8.0) and 0.03 mL of Ellman's reagent (4.0 mg/mL DTNB) in the same buffer.”
Comment 15: Paragraph 2.6.5: This is not an acceptable method’s description. How was the test conducted? What method was used for the scoring? Which aspects of the sensory profile were evaluated and how? How was the total score calculated?
Answers: Many thanks for your comment. We have revised the description of the methodology, the test methodology, the items tested and the calculation of the total score, details of which are shown in the revised Table 2. Modified description of the method in lines 235-249 in the revised manuscript which was shown as follows: “The sensory evaluation methods were based on the Roberta, et al. [31] methods with some modifications. According to the Chinese standard GB/T 31114-2014 (crite-rion for sensory evaluation of frozen drinks and ice cream), 20 reviewers were trained for 20 days to form the review panel. Samples were divided as separate portions in 30 mL plastic cups and were assessed in duplicate. The samples were coded and their or-der randomized. Ice cream samples were prepared and stored at −18 °C. Before evalu-ation, the samples were let at room temperature for 5 min. The samples were evaluated by the reviewers in turn for the following five main aspects: colour, morphology, or-ganization, taste and odour, impurity. Distilled water was required to eliminate the effects between samples after different flavours. A 100-point intensity scale (1-10 points for colour, 6-30 points for morphology, 5-30 points for organization, 5-20 points for taste and odour, and 3-10 points for impurity) was used for each term, as detailed in Table 2. Sensory acceptance was obtained by adding up the scores of colour, mor-phology, organization, taste and odour, and impurity.”
We have added the design used to conduct the sensory panel and the definitions of sensory terms into the body of the manuscript. Please see Table 2 in the revised manuscript.
Comment 16: Line 137: were instead of was.
Answers: Many thanks for your comment. We have revised “was” into “were” in line 147 in the revised manuscript.
Comment 17: Paragraph 3.3.1: It is important to state PPI’s theoretical isoelectric point value since you mention it and highlight its importance.
Answers: Many thanks for your comment. In this part, we are presenting a broad overview of the effect of the pH around isoelectric point on protein surface charges. Many proteins-related studies, such as whey protein, pea protein and soy protein, have confirmed the phenomenon and cited. Therefore, there is no way we can specifically emphasize what the isoelectric point of the PPI is. However, in the subsequent discussion of PPI isoelectric points, we have added specific isoelectric points in section 3.2.
Comment 18: Lines 275-276: Please rephrase, it is not at all clear.
Answers: Many thanks for your comment. We have rephrased in lines 305-306 in the revised manuscript which was shown as follows: “In summary, pH and heated can change the content of sulfhydryl bonds and affect the functional properties of proteins.”
Comment 19: Line 287: Micro-, not macro-
Answers: Many thanks for your comment. We have revised “macroparticulation” into “microparticulation” in line 323 in the revised manuscript.
Comment 20: Paragraph 3.1.3: It lacks explanation concerning the changes that were observed. For example, what does the increase of the β-sheet means for the protein? The results should be analyzed more extensively.
Answers: Many thanks for your suggestion. We have revised the discuss in lines 316-324 in the revised manuscript which was shown as follows: “The β-sheet content in PPM (pH 4.0) and PPM (pH 4.5) significantly increased to 41.64% and 39.73%, respectively; and the β-turn content had significantly decreased to 39.53% and 41.76%, respectively; random coil content showed no significant difference with PPI. The decreasing α-helix contents and increasing random coil content and β-sheet content were observed in PPM (pH 5.0) and PPM (pH 6.0). The secondary structures of PPM (pH 5.5) and PPM (pH 6.5) shifted from β-sheet and α-helix to β-turn and random coil, which is consistent with the results of Sun, et al. It can be concluded that the reduction in α-helical structures might have been due to that microparticulation destroyed the rigid structure of protein molecules, causing unfolding and rearrange. The rearranged protein molecules tend to have a higher degree of intermolecular flexibility.”
Comment 21: Paragraph 3.3.1: It could be useful to also stain the PPMs and observe the interphase through CLSM and compare the images to the emulsifying index.
Answers: Thank you very much for your suggestion. In our subsequent studies, we will stain the PPM, observe the interphase by CLSM and compare the images with the emulsifying index.
Comment 22: Paragraph 3.3.2: The results should be further analyzed and explain what they offer in understanding the new systems.
Answers: Many thanks for your comment. We have further analyzed and explained the rheological properties as follows: “In PPM (pH 7.0)-stabilized Pickering emulsion, the droplets are close enough together to interact with each other which may lead to the formation of a three-dimensional network of aggregated droplets. As the shear rate is increased, the hydrodynamic forces cause aggregates to become deformed and eventually disrupted which results in a reduction in the viscosity.” And the revised results and discussions were shown in lines 404-408 in the revised manuscript.
Comment 23: Line 475: “…which be owing to the different ice phase…” this is not a correct grammar type. Change it accordingly.
Answers: Many thanks for your comment. We have revised the sentence in lines 516-518 in the revised manuscript which was shown as follows: “However, the melting of PR100 (1.345%) was significantly greater than the other samples, due to the fact that the ice phase organization of PR100 is different from the other samples.”
Comment 24: Lines 475-477: The term “ice phase organization” should be explained and defined.
Answers: Many thanks for your comment. According to Kaleda, et al. [56], ice phase organization means the arrangement of ice phase. In order to be able to understand it better, we have revised “ice phase organization” into “ice phase arrangement” in lines 517 and 521 in the revised manuscript.
Comment 25: Lines 478-484: The results presented here contradict each other. How is it possible that while with the increase of substitution the melting also increases but that does not happen at 100% substitution?
Answers: Many thanks for your comment. We have revised the discuss in lines 516-519 in the revised manuscript which was shown as follows: “However, the melting of PR100 (1.345%) was significantly greater than the other samples, due to the fact that the ice phase arrangement of PR100 is different from the other samples. The PR100 with lower overrun and stability there was not enough network space structure to prevent melting and complete collapse.”
Comment 26: Lines 495-501: Could you propose a way to avoid those defects?
Answers: Many thanks for your comment. A moderate amount of substitution with PPM(pH 7.0)-stabilized Pickering emulsion can compensate for such structural defects. When the substitution rate is below 60%, it can be appropriate to compensate for the texture and ice crystal problems. However, when the substitution rate is higher than 80%, structural defects cannot be avoided.
Comment 27: Conclusions should highlight more the major findings of this study and advocate for its novelty and possible applications in the food industry in general. Also, it should be grammar proofed.
Answers: Many thanks for your comment. We have revised the conclusions in line 550-560 in the revised manuscript which was shown as follows: “PPM (pH 7.0) was suitable for application in O/W emulsion stabilization because of its proper particle size, more flexible structure, high EAI and ESI. The Pickering emulsion stabilized by PPM (pH 7.0) had uniform oil droplet distribution and similar rheological properties with cream, so can be used as saturated fat substitutes in the production of ice cream. PPM (pH 7.0)-stabilized emulsion can be used to substitute 60% cream to produce low-saturated fat ice cream which had high structural stability, sim-ilar melting properties, overrun and sensory properties with PR0. The results indicated that PPM (pH 7.0)-stabilized Pickering emulsion has the fatty acid profile of the corn oil used, but he rheological properties similar to saturated fatty acid. Therefore, it is feasible to be used as saturated fat substitute in the production of low saturated fat food to meet the demand of consumers for unsaturated fat.”
Reviewer 2 Report
Comments and Suggestions for Authors
The manuscript titled “Fabricating pea protein microgel-stabilized Pickering emulsion as saturated fat reduction in ice cream” is interesting and important because the consequences of diet-related diseases are very serious, they will significantly affect the quality of life. Unfortunately, the manuscript requires refinement. Comments and suggestions are included below.
Line 107: „acid” instead „ac-id”
Line 111-112: Which solutions were used to obtain the desired pH values? What was the initial pH of the PPI dispersion?
Line 116: I suggest 8000 rpm. Please revise the entire manuscript.
Line 121: Please complete the methodology:
What was the scattering angle?
At what temperature were the measurements performed?
How were the samples prepared for measurements?
Line 132: Please use Fourier-transform infrared spectroscopy in the headline.
Line 134: Please write the units correctly.
Line 162: “… were measured using a clamp with a conical (25 mm) diameter and a distance of 1 mm between parallel plates.” Please explain what measurement system was used. What was the temperature of the measurements?
Line 167: Please write the units correctly.
Line 169: Please complete the measurement time.
Line 172: Please write the units correctly.
Line 210: Please provide the conditions for ice cream production and the characteristics of the device.
Line 214: section 2.5.4. It is not number of rheological properties’ section.
Line 224: Why was the sensory assessment performed by students and not by a specialized sensory panel? What rating scale was used? What is hidden under the statement: other indicators?
Figure 1 and line 238-244: In section 2.3.1. Particle size distribution of PPMs, indicated that the measurements were used using Nano Brook 90Plus particle size analyzer (Brookhaven, Massachusetts, USA). article size of PPMs are outside the measuring range of the instrument. Please explain.
Figure 1( A,B): Significant differences or their absence should be marked for values arranged in a non-decreasing series
Figure 1 (B): Please write the unit correctly. If there are such large differences in the obtained values, I suggest using a different type of chart than a bar chart and using the appropriate scale of the y-axis.
Line 451-452: “G' and G" of all ice cream slurries except PR40 and PR60 increased with increasing saturated fat substitution rate until 1.3 Hz, which indicated the destruction of the structures.” This cannot be seen in the graphs drawn. Not only in the case of the two systems indicated.
Figure 6: the areas and points marked in the drawing cover the area of non-linear viscoelasticity, the interpretation of which requires appropriate description and in-depth analysis using mathematical methods. The course of the curves is characteristic of certain types of systems and results from the components used. Please correct the discussion.
Table 3: How was the result of the sensory analysis obtained/calculated? Please explain, and add complete information to manuscript.
Author Response
Reviewer 2:
Comment: The manuscript titled “Fabricating pea protein microgel-stabilized Pickering emulsion as saturated fat substitutes in ice cream” is interesting and important because the consequences of diet-related diseases are very serious, they will significantly affect the quality of life. Unfortunately, the manuscript requires refinement. Comments and suggestions are included below.
Answers: We appreciate the selfless contributions to this manuscript from you. It is because of your insightful and helpful comments and suggestions, we gained the confidence to improve our work better. The language errors in the manuscript foods-2985301 entitled “Fabricating pea protein microgel-stabilized Pickering emulsion as saturated fat substitutes in ice cream” have been carefully revised with the revisions marked in red.
Comment 1: Line 107: “acid” instead “ac-id”
Answers: We have revised “ac-id” into “acid” in line 108 in the revised manuscript.
Comment 2: Line 111-112: Which solutions were used to obtain the desired pH values? What was the initial pH of the PPI dispersion?
Answers: Many thanks for your comment. We have revised in line 112-114 in the revised manuscript which was shown as follows: “PPI (18 g/100 mL) dispersion was magnetically stirred for 2 hours, and the initial pH of PPI dispersion was 7.22. Then, the pH of PPI dispersion was adjusted from 4.0 to 7.0 using 1 M HCl solution.”
Comment 3: Line 116: I suggest 8000 rpm. Please revise the entire manuscript.
Answers: Many thanks for the referee’s comment. We have replaced “r/min” with “rpm” in line 136 and the entire revised manuscript.
Comment 4: Line 121: Please complete the methodology:
What was the scattering angle?
At what temperature were the measurements performed?
How were the samples prepared for measurements?
Answers: Many thanks for your comment. We have added information to complete the methodology which was shown in lines 123-130 in the revised manuscript and shown as follows: “Particle size distribution of the protein micro-gel was measured by Battersize2000 particle size analyzer (Better, Liaoning, China). The samples were added to a stirred measuring cell containing 800 mL of deionized water at room temperature, with deionized water as the dispersing medium, and the light shading percentage should be 10% to 15% in order to avoid the multiple scattering effect of proteins and to ensure the accuracy of the measurement. The relative refractive indices of the microgel particles and the dispersing medium (deionized water) were 1.52 and 1.33, respectively. Each sample with different treatments was measured three times and the average value was taken.”
Comment 5: Line 132: Please use Fourier-transform infrared spectroscopy in the headline.
Answers: Many thanks for your comment. We have revised“FTIR” into “Fourier-transform infrared spectroscopy of PPI and PPMs” in line 142 and 306 in the revised manuscript.
Comment 6: Line 134: Please write the units correctly.
Answers: Many thanks for your comment. We have revised the units “4000-400 cm-1, 4 cm-1” into “ 4000-400 cm-1, 4 cm-1′′ ” in line 144 in the revised manuscript.
Comment 7: Line 162: “… were measured using a clamp with a conical (25 mm) diameter and a distance of 1 mm between parallel plates.” Please explain what measurement system was used. What was the temperature of the measurements?
Answers: Many thanks for your comment. We have revised in lines 170-171 in the revised manuscript which was shown as follows: “The emulsion was characterized by MARS 60 rheometer (Haake, Karlsruhe, Germany) at 25°C”.
Comment 8: Line 167: Please write the units correctly.
Answers: Many thanks for your comment. We have revised the units “ Pa.sn′, Pa.sn′′ ” into “ Pa.sn′, Pa.sn′′ ” in line 177 in the revised manuscript.
Comment 9: Line 169: Please complete the measurement time.
Answers: Many thanks for your comment. We have completed the measurement time in line 182 in the revised manuscript which was shown as follows: “Shear rate was set to increase from 1 s-1 to 300 s-1 and maintaining in 400 s.”
Comment 10: Line 172: Please write the units correctly.
Answers: Many thanks for your comment. We have revised the units “γ, s-1” into “ γ, s-1” in line 182 in the revised manuscript.
Comment 11: Line 210: Please provide the conditions for ice cream production and the characteristics of the device.
Answers: Many thanks for your comment. We have added information on ice cream production conditions and equipment characteristics in lines 215-221 in the revised manuscript, as follows: “Different substitution rates (0%, 20%, 40%, 60%, 80% and 100%) of cream in ice cream were substituted by PPM(pH 7.0)-stabilized emulsion (Table 1). And the sam-ples were named PR0, PR20, PR40, PR60, PR80 and PR100. They were homogenized for 5 min at 3000 rpm after mixing and then were pasteurized at 72 ℃ for 30 min. They were cooled and stored at 4 ℃ for 4 h. Finally, the finished product was gotten after pouring them into an ice cream maker (Pink Bunny, Guangdong, China). Three parallel experiments for each sample were carried out.”
Comment 12: Line 214: section 2.5.4. It is not number of rheological properties’ section.
Answers: Many thanks for your comment. We have revised "section 2.5.4." into " section 2.5.3" in line 225 in the revised manuscript.
Comment 13: Line 224: Why was the sensory assessment performed by students and not by a specialized sensory panel? What rating scale was used? What is hidden under the statement: other indicators?
Answers: Many thanks for your comment. We have revised the description of the methodology, details of which are shown in the revised Table 2. Modified description of the method in lines 235-249 in the revised manuscript which was shown as follows: “The sensory evaluation methods were based on the Roberta, et al. [31] methods with some modifications. According to the Chinese standard GB/T 31114-2014 (criterion for sensory evaluation of frozen drinks and ice cream), 20 reviewers were trained for 20 days to form the review panel. Samples were divided as separate portions in 30 mL plastic cups and were assessed in duplicate. The samples were coded and their order randomized. Ice cream samples were prepared and stored at −18 °C. Before evaluation, the samples were let at room temperature for 5 min. The samples were evaluated by the reviewers in turn for the following five main aspects: colour, morphology, organization, taste and odour, impurity. Distilled water was required to eliminate the effects between samples after different flavours. A 100-point intensity scale (1-10 points for colour, 6-30 points for morphology, 5-30 points for organization, 5-20 points for taste and odour, and 3-10 points for impurity) was used for each term, as detailed in Table 2. Sensory acceptance was obtained by adding up the scores of colour, morphology, organization, taste and odour, and impurity.”
We have added the design used to conduct the sensory panel and the definitions of sensory terms into the manuscript. Please see revised Table 2 in the revised manuscript.
Comment 14: Figure 1 and line 238-244: In section 2.3.1. Particle size distribution of PPMs, indicated that the measurements were used using Nano Brook 90Plus particle size analyzer (Brookhaven, Massachusetts, USA). Particle sizes of PPMs are outside the measuring range of the instrument. Please explain.
Answers: Many thanks for your comment. We are sorry for the mistake of the particle sizer machine and have revised it in line 123-130 in the revised manuscript which was shown as follows: “Particle size distribution of the protein micro-gel was measured by Battersize2000 particle size analyzer (Better, Liaoning, China). The samples were added to a stirred measuring cell containing 800 mL of deionized water at room temperature, with de-ionized water as the dispersing medium. And the light shading percentage should be 10% to 15% in order to avoid the multiple scattering effects of proteins and to ensure the accuracy of the measurement. The relative refractive indices of the microgel parti-cles and the dispersing medium (deionized water) were 1.52 and 1.33, respectively. Each sample with different treatments was measured three times and the average value was taken.”
Comment 15: Figure 1(A, B): Significant differences or their absence should be marked for values arranged in a non-decreasing series.
Answers: Many thanks for your comment. We have added the absence remarks on significance marking in lines 279-280 in the revised manuscript which was shown as follows: “The values that do not bear the same letter on the column are significantly different(p<0.05).”
Comment 16: Figure 1 (B): Please write the unit correctly. If there are such large differences in the obtained values, I suggest using a different type of chart than a bar chart and using the appropriate scale of the y-axis.
Answers: Many thanks for your comment. We have changed the correct units and adjusted the scale of y-axis. Please see revised Figure 1B in the revised manuscript. We referenced a number of papers and found that bar charts were the most appropriate form of presentation.
Yang, J.; Duan, Y.; Zhang, H.; Huang, F.; Wan, C.; Cheng, C.; Wang, L.; Peng, D.; Deng, Q. Ultrasound coupled with weak alkali cycling-induced exchange of free sulfhydryl-disulfide bond for remodeling interfacial flexibility of flaxseed protein isolates. Food Hydrocolloids 2023, 140, 108597, https://doi.org/10.1016/j.foodhyd.2023.108597.
Yang, C.; Liu, J.; Han, Y.; Wang, B.; Liu, Z.; Hu, H.; Guan, Z.; Yang, Y.; Wang, J. Fabrication of polyphenol-pumpkin seed protein isolate (PSPI) covalent conjugate microparticles to protect free radical scavenging activity of polyphenol. Food Bioscience 2023, 55, doi:10.1016/j.fbio.2023.102982.
Comment 17: Line 451-452: “G' and G" of all ice cream slurries except PR40 and PR60 increased with increasing saturated fat substitution rate until 1.3 Hz, which indicated the destruction of the structures.” This cannot be seen in the graphs drawn. Not only in the case of the two systems indicated.
Answers: Many thanks for your comment. We have revised the sentence in lines 492-496 in the revised manuscript as follows: “G' and G" of all ice cream slurries except PR40 and PR60 increased with increasing frequency until 1.3 Hz, which indicated the destruction of the structures.”
Comment 18: Figure 6: the areas and points marked in the drawing cover the area of non-linear viscoelasticity, the interpretation of which requires appropriate description and in-depth analysis using mathematical methods. The course of the curves is characteristic of certain types of systems and results from the components used. Please correct the discussion.
Answers: Many thanks for your comment. When the sample structure remains stable, both G' and G” gradually increase with increasing frequency. However, when the frequency increases to a value where the sample structure breaks down, the values of G' and G” decrease significantly. This frequency is indicative of the structural stability of the sample. It is also the point labeled in Figure 6. We have already revised the discussion in lines 492-496 in the revised manuscript as follows: “G' and G" of all ice cream slurries except PR40 and PR60 increased with increasing frequency until 1.3 Hz, which indicated the destruction of the structures. G’ and G’’ of PR40 and PR60 remained stable until 2.8 Hz, indicating that the structure is still intact. This also indicates the enhanced structural stability of low-fat ice cream slurries with 40% and 60% substitution rates.”
Comment 19: Table 3: How was the result of the sensory analysis obtained/calculated? Please explain, and add complete information to manuscript.
Answers: Many thanks for your comment. Sensory acceptance was obtained by adding up the scores of colour, morphology, organization, taste and odour, and impurity. We have revised the description of the methodology, the test methodology, the items tested and the calculation of the total score, details of which are shown in the revised Table 2. Modified description of the method in lines 235-249 in the revised manuscript which was shown as follows: “The sensory evaluation methods were based on the Roberta, et al. [31] methods with some modifications. According to the Chinese standard GB/T 31114-2014 (criterion for sensory evaluation of frozen drinks and ice cream), 20 reviewers were trained for 20 days to form the review panel. Samples were divided as separate portions in 30 mL plastic cups and were assessed in duplicate. The samples were coded and their order randomized. Ice cream samples were prepared and stored at −18 °C. Before evaluation, the samples were let at room temperature for 5 min. The samples were evaluated by the reviewers in turn for the following five main aspects: colour, morphology, organization, taste and odour, impurity. Distilled water was required to eliminate the effects between samples after different flavours. A 100-point intensity scale (1-10 points for colour, 6-30 points for morphology, 5-30 points for organization, 5-20 points for taste and odour, and 3-10 points for impurity) was used for each term, as detailed in Table 2. Sensory acceptance was obtained by adding up the scores of colour, morphology, organization, taste and odour, and impurity.”
We have added the design used to conduct the sensory panel and the definitions of sensory terms into the manuscript. Please see Table 2 in the revised manuscript.